

# Agricultural use of compost under different irrigation strategies in a hedgerow olive grove under Mediterranean conditions. A comparison with traditional systems

Laura L. de Sosa[1], María Jose Martín-Palomo[2,3], Pedro Castro-Valdecantos[2,3], Engracia Madejón[1]

[1] Instituto de Recursos Naturales y Agrobiología de Sevilla (IRNAS-CSIC) Av. Reina Mercedes 10 41012 Sevilla
[2] Dpto. Ciencias Agroforestales, ETSIA, Universidad de Sevilla, Crta de Utrera Km 1, E-41013 Sevilla, Spain
[3] Unidad Asociada al CSIC de Uso Sostenible del Suelo y el Agua en la Agricultura (US-IRNAS), Crta de Utrera Km 1, E-41013, Sevilla, Spain

*Correspondence to*: Laura L. de Sosa (lauralozano@irnsa.csic.es)

**Abstract.** Soil and water efficient management are key factors to ensure olive sustainable production. The use of compost based on olive waste (alperujo) as fertilizer could enhance ecosystem services while the need to transition to a zero waste circular economy is achieved. The present work includes a comparative study of the effect of alperujo compost (AC) vs inorganic fertilization under different management systems: an intensive traditional adult olive grove under rainfed conditions and a young hedgerow olive system, in which a factorial test of tree irrigation regimes (full, deficit and no irrigation) is implemented as well. At the hedgerow plots, the addition of AC and soil sampling time greatly impacted soil chemical parameters and to a lesser extent, enzymatic activities whereas irrigation regimes did not exert a mark influence. In the traditional rainfed system, the addition of AC proved to be an efficient tool for carbon sequestration. The first soil sampling revealed a clear stoichiometric relationship between soil organic matter (SOM) and the NPK contents at both systems whereas the correlations were weak and scarce in the second sampling at the hedgerow plots. This fact was related to a decay of the compost effect. Compost in combination with irrigation tended to trigger a certain priming effect on the native SOM with time since the carbon stocks were reduced between 6-38% from one sampling to the other in the hedgerow system depending on the irrigation intensity. However, the deficit irrigation caused a less intense reduction of the SOM and essential nutrients representing the best alternative to maximize the agronomics effects of the compost under a water-saving strategy. Recurrent application of compost would be necessary to maintain soil quality, especially with high tree densities. The combined management of AC and the deficit irrigation proved to be an efficient tool toward a zero waste circular economy and a water conservation strategy.

## 1 Introduction

Olive-growing is a pivotal piece in Mediterranean countries as an economic engine and a source of employment due to its production under adverse conditions, adaptability, and its natural abundance (Kostelenos and Kiritsakis, 2017). Spain, one of the leading producers, accounts for 2.62 million hectares which represent 20% of the world's olive area (FAO, 2021). Over



the last decades, olive cultivation has experienced a fast and large-scale intensification process in areas characterized by
medium to poor soil fertility and water scarcity (Kavvadias and Koubouris, 2019). Among crops, olive trees have a long
tradition of high tolerance to low soil fertility and hydric stress (Ahumada-Orellana et al., 2017). However, these strong
changes in olive grove structures (e.g., higher densities and hedgerow formation) are leading to a great deal of pressure on
these ecosystems where the control of irrigation and soil resources has become essential to ensure its sustainability and
production (García-Garví et al., 2022; Morgado et al., 2022).

New sustainable strategies are needed to adapt an efficient water use and fertilization according to the phenological tree state
and its nutritional needs (Cano-Lamadrid et al., 2015). In fact, there is evidence that suggest that this tree phenology could be
altered by a differential development related to the use of the organic amendment as fertilizers (Mekki et al., 2019).
Consequently, the new European strategies foster and fund sustainable management practices based on increasing soil organic
carbon inputs and optimizing water resources to maintain long-term productivity and preserve agroecosystem services
(Hernández et al., 2015). In this sense, the use of by-products from olive oil production (e.g., solid by-products of the two-
phase centrifugation method for olive oil extraction called alperujo) is gaining more attention as it constitutes an interesting
and sustainable option as a source of nutrients and carbon (Calvano and Tamborrino, 2022; de Sosa et al., 2022). Alperujo
which is one of the major wastes from the oil industry is of particular interest due to its high water percentage, high salt
concentration, and organic matter (Ghilardi et al., 2022). Its agricultural use allows an effective valorization of a widely spread
agricultural waste into a value-added product with nutritional benefits for the soil, reducing in turn the need for inorganic
fertilizers. Studies have shown that amendments based on alperujo can influence the soil enzymatic activity that controls the
patterns of organic matter decomposition (Panettieri et al., 2022), increase soil fertility through the slow release of nutrients
(Alburquerque et al., 2011), improve soil water–soluble carbon (Madejón et al., 2016), modify soil chemical properties
(Podgornik et al., 2022) or the oil quality (Proietti et al., 2015). In this sense, irrigation management has been proposed as a
central piece to maximize the effect of the organic amendments and as a tool to palliate the effect of climate change (Kavvadias
and Koubouris, 2019; Mairech et al., 2021; Michalopoulos et al., 2020). However, little attention has been paid to how the
agronomic effects of the compost or organic amendments can evolve under different irrigation regimes, essential with high
tree densities. Hirich et al., (2014) showed how the combined effect of deficit irrigation (in contrast to full irrigation) and the
organic amendment was able to maximized yield and biomass production while Baghbani-Arani et al. (2020) observed better
water use efficiency and an increase in sunflower productivity when an organic amendment was applied regardless of the
irrigation regime. Kavvadias and Koubouris (2019) also showed that most of the soil properties were favoured by irrigation.
However, this combined effect of the organic amendment and irrigation has a number of limitations that need to be considered.
Thus, research in this field has showed some disagreement on the negative (and positive) effects of the compost due to the
variability in experimental conditions (e.g., the dose used, phenological stage of the crop, climatic conditions, or spreading
methods) (Regni et al., 2017). Moreover, there are some major drawbacks of irrigation such as poor irrigation water quality,
water overuse, exacerbation of soil erosion and leaching of soil   nutrients (Khaliq and Kaleem Abbasi, 2015; Morgado et al.,
2022; Sousa et al., 2019). Consequently, studies that clarify the relationships between nutrition and water application are of
great interest to maximize resources under a climate change scenario.

The present work was aimed at (i) evaluating the effect of alperujo compost (AC) on soil physical-chemical properties and
enzyme activities under two different management systems (hedgerow/intensive olive groves) after repeated AC applications
(ii) verifying the feasibility and sustainability of the AC as fertilizers under high tree densities and different irrigation regimes.
We hypothesize that the supply of stabilized organic matter could alleviate the negative effects of water stress while improving
water efficiency and soil fertility. To that end, we evaluated changes in some key soil physical, chemical and biochemical
parameters as indicators of soil quality.

## 2 Methods

### 2.1 Experimental area and experimental design





A study was carried out at the agriculture experimental farm ''La Hampa'' of the ''Instituto de Recursos Naturales y Agrobiología de Sevilla (IRNAS-CSIC)'' (37°17'01.8"N 6°03'57.4"W). The soil was a calcic Cambisol (IUSS Working Group WRB, 2015) characterized by a sandy clay loam texture, low fertility, and low organic matter content (pH: 7.5; TOC: 8 g kg-80 1; N: 0.8 g kg-1; Olsen P: 10 mg kg-1; Available-K: 200 mg kg-1). The climate is typically Mediterranean, with 3–5 months of summer droughts and moderately wet cool winters. A summary of the meteorological data of the experimental area can be found in Figure S1.

The experiment was conducted between December 2020 and June 2022 in two olive grove areas with different managing strategies. The first experimental site compromises an area of 0.7 ha of a young olive grove of cultivar Manzanilla, planted in 85 a pattern of 4 m × 1.5 m as a hedgerow system established in 2018. The area was divided into 18 plots (ca. 410 m² consisting of 5 lines of trees, 1.666 trees ha⁻¹) and had a completely randomized design with irrigation and fertilization as the main experimental factors as explained in Figure 1. The second study site was set in an area of 1.2 ha of an adult rainfed olive grove of the cv Manzanilla, planted in a pattern of 7 m × 5 m with intensive traditional management established in 1997. The study site was divided into 20 plots (ca. 400 m², 285 trees ha⁻¹) of which 8 were selected to carry out the present experiment (Fig. 1).


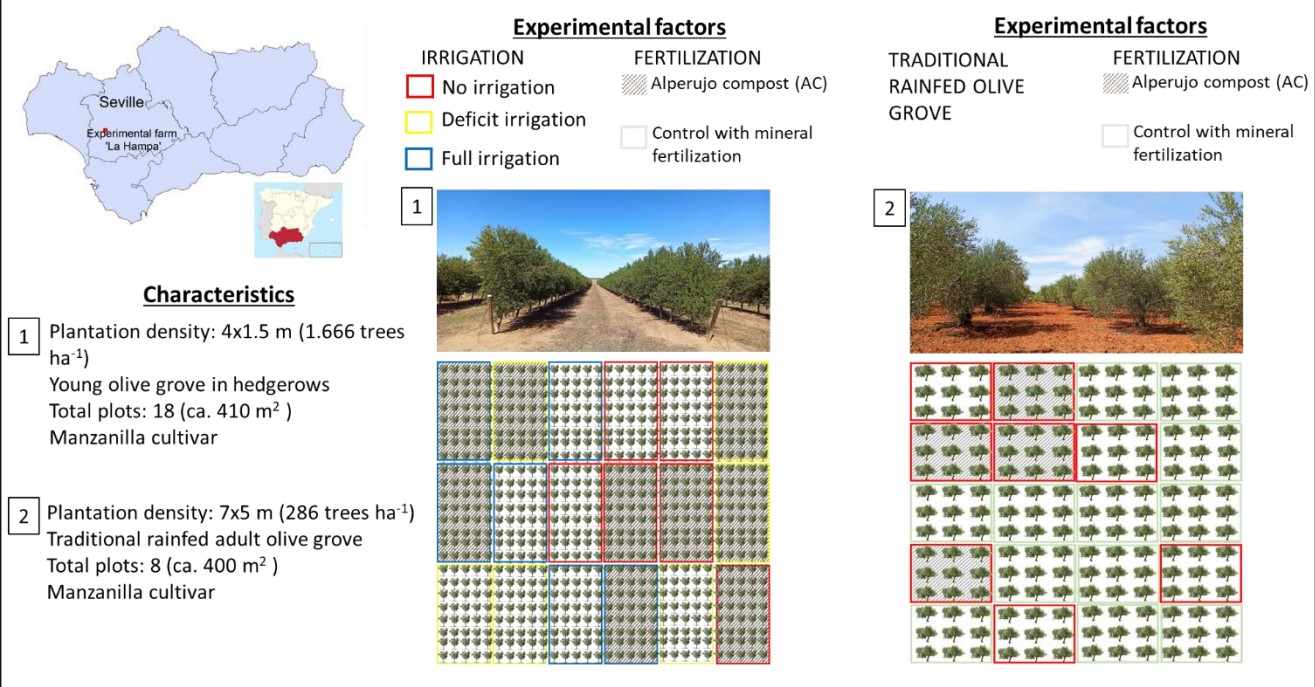

**Figure 1** Experimental plots in two olive grove areas with different management strategies at the experimental farm 'La Hampa' located in Coria del Río, Seville (Spain).




### 2.1.1 Fertilization

In December 2020, AC, the compost based on solid waste from two-phase olive oil extraction (alperujo), was applied in the intensive traditional plots at a rate of 17 t ha-1. This compost was made of 60% alperujo and 40% pruning wastes and legumes. A detailed description of the main parameters of the AC can be found in Table 1. The product was supplied by an olive oil cooperative after a composting process for more than 12 months. The same product with the same dose was applied in the hedgerow plots in July 2021 to test the short-lasting effect of the organic amendment under high densities. Additionally, trees of all treatments were supplemented with three foliar applications of $KNO_3$ at a rate of 12.5 kg ha$^{-1}$ each time and one application of B (2 l ha$^{-1}$) before fruit set.

**Table 1** Characterization of the alperujo compost (AC) used in this study. Data are mean values ($n = 3$) $\pm$ standard error of the mean (SEM).



| Parameter | Average Value |
|---|---|
| Humidity (%) | $21.6 \pm 0.33$ |
| pH | $9.19 \pm 0.01$ |
| EC (mS cm$^{-1}$) | $5.88 \pm 0.24$ |
| OM (%) | $58.4 \pm 1.34$ |
| N (%) | $2.50 \pm 0.26$ |
| $NH_4^+$-N | $56.8 \pm 2.69$ |
| $NO_3^-$-N | $26.1 \pm 0.78$ |
| $P_2O_5$ (%) | $2.52 \pm 0.21$ |
| $K_2O$ (%) | $3.38 \pm 0.07$ |
| CaO (%) | $9.40 \pm 1.04$ |
| Mg O (%) | $2.34 \pm 0.01$ |
| Na (%) | $0.93 \pm 0.003$ |
| $SO_3$ (%) | $2.14 \pm 0.06$ |
| Fe (g kg$^{-1}$) | $5.42 \pm 0.94$ |
| Mn (mg kg$^{-1}$) | $332 \pm 34.9$ |
| Cu (mg kg$^{-1}$) | $69.5 \pm 7.83$ |
| Zn (mg kg$^{-1}$) | $301 \pm 21.3$ |
| B (mg kg$^{-1}$) | $52.9 \pm 0.92$ |
| As (mg kg$^{-1}$) | $<0.10 \pm 0.000$ |
| Cd (mg kg$^{-1}$) | $0.09 \pm 0.03$ |
| Co (mg kg$^{-1}$) | $1.70 \pm 0.11$ |
| Cr (mg kg$^{-1}$) | $38.6 \pm 2.65$ |
| Ni (mg kg$^{-1}$) | $19.0 \pm 1.06$ |
| Pb (mg kg$^{-1}$) | $5.47 \pm 0.64$ |

In March 2022, the following treatments were applied to each site:

(1)    AC at a rate of 17 t ha-1 in plots with organic fertilization in both sites

(2)    nitrofoska perfect (15-5-20) at a rate of 1.7 kg/árbol at the intensive site and 63 g/árbol at the hedgerow plots in plots
with inorganic fertilization

 (3) fertirrigation at a variable rate (Table 2) for the treatments of full and deficit irrigation

As in 2021, trees of all treatments were supplemented with three foliar applications of KNO3 at a rate of 12.5 kg ha-1 each
time and one application of B (2 l ha-1) before fruit set.





Phytosanitary treatments consisted of the application of Cu as a fungicide and two applications of dimethoate as an insecticide.

**Table 2** Nutrient concentrations (kg/ha) calculated depending on the crop demands. The irrigation dose was adapted to deliver the same nutrient concentration adjusting the amount of water for the full and deficit treatments.

| Kg ha$^{-1}$ | April | May | June | July | August | Sept | Oct |
|---|---|---|---|---|---|---|---|
| N | 11 | 24 | 24 | 24 | 11 | 11 | 5 |
| P | 2.5 | 5 | 5 | 5 | 5 | 5 | 2.5 |
| K | 5 | 13 | 13 | 27 | 27 | 27 | 13 |

**2.1.2 Irrigation regimens**

Three irrigation treatments were selected for the hedgerow plots: (1) full irrigation, (2) deficit irrigation and (3) no irrigation. The trees water status was characterised with stem water potential ($\Psi$) and leaf conductance. The water potential was measured at midday in one leaf per tree, using the pressure chamber technique (Scholander et al., 1965). The leaves near the main trunk were covered in aluminium foil at least two hours before measurements were taken every 7-10 days. Leaf conductance was

measured at midday in the same trees that water potential with a dynamic diffusion porometer (DC-1, Decagon, UK). Irrigation was carried out during the night by drip, using one lateral pipe per row of trees and three emitters per plant, delivering 2 L h$^{-1}$ each. All the measurements were made on the central tree of each plot.

(1) **Full irrigation** (F): when the irrigation schedule is programmed to supply the 100% of the crop evapotranspiration (ETc). This water dose was increased to 125% ETc if the water potential measurements were more negative than those estimated by

the baseline established in Corell et al. (2016). Same irrigation treatment combined with the compost addition is stated as FC.

(2) **Deficit irrigation** (D): when conditions of low or moderate stress are maintained during several phenological stages. The water dose was 1mm/day along the irrigation season. This applied water was changed accordingly to the water status and phenological stages of the trees. During all the seasons, except the pit hardening period, from mid-June to the end of August, water potential was compared with Corell et al. (2016)'s baseline. Applied water was increased (in 1, 2, 3 mm) when measured

values were more negative than expected (10%, 20% 30% more negative). During the pit hardening, the threshold value decreased until -2 MPa according to Girón et al. (2015). Same irrigation treatment combined with the compost addition is stated as DC.

(3) **No irrigation** (NI): no irrigation was applied from even under conditions of severe water stress. Same irrigation treatment combined with the compost addition is stated as NIC.

A detailed description of the amount of water provided monthly according to the irrigation treatment can be found in Table S1.

An intensive traditional rainfed regime without supplemental irrigation was implemented at the intensive traditional plots.

**2.2 Soil sampling, chemical analysis and enzyme activities analysis**





Three soil cores (0-10 cm) per plot were taken and merged together to obtain a composite sample per plot in November 2021
(after the first compost application) and April 2022 (after the second compost application) in both experimental sites. After
sieving at 2 mm, soil samples were split into two subsamples: the first one was stored at 4 °C prior to enzymatic analysis in
laboratory; the second one was air for chemical analysis.

Sample dry weights were used to calculate soil water content (SWC) by the gravimetric method. Soil pH and electrical

conductivity (EC) were measured in the water extract (1:5, m/v) after shaking for 1 h using a pH meter (CRISON micro pH
2002), respectively. Soil organic matter (SOM) was calculated by dichromate oxidation and titration with ferrous ammonium
sulphate (Walkley and Black, 1934). Water-soluble carbon (WSC) content was determined using a TOC-VE Shimadzu
analyzer after extraction with water using a sample-to-extractant ratio of 1:10. Total Kjeldahl-N (TN) was determined by the
method described by Hesse (1971). Nitrate ($NO_3^-$-N) was extracted in water (1:5 w/v) and quantified in the aqueous extracts

by a continuous flow auto-analyser Luebbe GmbH AA3 dual channel (Norderstedt, Germany). Ammonium ($NH_4^+$-N) was
extracted in KCl 2 M (1:5 w/v) and determined using the same flow auto-analyzer. Available-P was determined after extraction
with sodium bicarbonate at pH 8.5 (Olsen et al., 1954), while available-K was determined after extraction with ammonium
acetate at pH 7.5 (Freitas, 1970). Dehydrogenase activity (DHA) was determined according to Trevors (1984) after soil
incubation with p-iodo nitrotetrazolium chloride (INT) and measurement of the p-iodo-nitrotetrazidin formazan (INTF)

absorbance at 490 nm. Glucosidase activity (β-Glu) was measured as indicated by Eivazi and Tabatabai (1988) after soil
incubation with p-nitrophenyl-β-D-glucopyranoside and measurement of the p-nitrophenol absorbance at 400 nm.. Urease
activity was determined according the method proposed by Kandeler and Gerber (1988) and modified by Kandeler et al. 1999.
A chronogram of the compost addition, soil sampling and irrigation months performed during the experiment is displayed in
Figure S1.


### 2.3 Statistical analysis

A multi-way ANOVA followed by Tukey's post-hoc test was performed for the hedgerow plots to test the effect of compost
addition, irrigation and sampling time on soil physico-chemical parameters and enzymatic activities and a two-way ANOVA
with compost addition and sampling time as main factors was performed in the intensive plots. For all statistical tests, $p < 0.05$

was selected as the significance cut-off value. Statistical analysis was performed with SPSS v25 for Windows (IBM Corp.,
Armonk, NY).  Heatmaps of pearson's correlation coefficients was performed with Origin Pro v2022 software.

### 3. Results

### 3.1 Main drivers controlling soil physical and chemical parameters in a hedgerow grove with irrigation

Compost addition and sampling time exerted the greatest influenced on soil chemical parameters and to a lesser extent
irrigation (Table 3 and S2). The first compost addition increased SWC in 33% for the DC and 6% for the FC treatment
compared to their respective control, whereas no changes were detected for the NIC treatment and its control. The second
compost addition improved SWC in 13% and 26% for NIC and DC compared to their controls respectively.



**Table 3** Results of ANOVA (F and *p*-value) showing the main significant factors (i.e. compost addition, irrigation, sampling time or interactions) controlling changes in soil physical and chemical properties at the hedgerow plots. SWC: soil water content, EC: electrical conductivity, SOM: soil organic matter, WSC: water-soluble carbon, Avail-K: available K.

| ANOVA results | SWC | | pH | | EC | | SOM | | WSC | | TN | | NO$_3^-$-N | | NH$_4^+$-N | | Olsen-P | | Avail-K | |
|---|---|---|---|---|---|---|---|---|---|---|---|---|---|---|---|---|---|---|---|---|
| | F | *p* | F | *p* | F | *p* | F | *p* | F | *p* | F | *p* | F | *p* | F | *p* | F | *p* | F | *p* |
| Compost addition | 7.58 | * | ns | | ns | | 22.2 | *** | 53.0 | *** | 12.6 | ** | ns | | ns | | 18.7 | *** | 28.2 | *** |
| Irrigation | 15.6 | *** | ns | | 8.28 | * | ns | | 7.17 | ** | ns | | ns | | ns | | ns | | ns | |
| Sampling time | 17.1 | *** | 49.0 | *** | 25.2 | *** | 14.9 | *** | 58.6 | *** | ns | | 13.0 | *** | 54.2 | *** | ns | | ns | |
| Compost*irrigation* sampling time | ns | | ns | | ns | | ns | | 3.77 | * | ns | | ns | | ns | | ns | | ns | |
| Compost*sampling time | ns | | ns | | ns | | 6.10 | * | 7.17 | * | ns | | ns | | 5.33 | * | ns | | ns | |
| Irrigation*sampling time | ns | | 5.70 | ** | ns | | ns | | ns | | ns | | ns | | ns | | ns | | ns | |
| Compost*irrigation | ns | | ns | | 3.92 | * | ns | | ns | | ns | | ns | | ns | | ns | | ns | |

Significance level: *$p$<0.05; **$p$<0.01, *** $p$<0.001

Values of pH were highly sensitive to seasonal changes and to a lesser degree, to the irrigation management but not a clear trend regarding the latter was observed during the second sampling time.

Values of EC increased with the addition of compost and under a higher irrigation dose (DC, F and FC) (Table S2). This effect tended to disappear over time (Table 3 and S2).

Compost addition increased SOM on average a 28% regardless the irrigation treatment at the first sampling time whereas an overall increment of 12% after the compost addition and irrespective of the irrigation dose was detected during the second sampling (Table 3 and S2). The amount of WSC was influenced by the combination of the three factors (i.e. compost, sampling time and irrigation) showing an increase of 34% for the NIC and DC treatments and 54% for the FC during the first sampling while the percent improvement over their control was 34%, 58% and 18% for NIC, DC, FC respectively for the second sampling. Compost had a significant positive effect in TN, P and K concentration (p < 0.01). Thus, TN increased on average by 23%, P in 46% and K in 41% with the addition of compost and irrespectively of sampling time, and irrigation regime.

No significant changes were detected with respect to NO3 ⁻-N and NH4+-N contents but some trends were identified. During the first sampling, the compost addition increased by 26% and 54% NO3⁻-N content in NIC and DC treatments whereas the FC treatment caused a decreased of 74% in NO3⁻-N content in comparison with its control. The content of NH4+-N increased on average with AC addition irrespective of the irrigation regime. Both inorganic forms were greatly reduced in the second sampling for all treatments.



**3.2 Main drivers controlling soil physical and chemical parameters with an intensive traditional management with no irrigation**

As the hedgerow plots, the addition of compost significantly impacted some of the main soil chemical parameters but in a
lesser extent. The SWC differed by 20% between the plots with AC addition and their control irrespective of the sampling time although this difference was not significant.

Values of soil pH increased by 0,6-0,8 units on average with the addition of AC and the same trend was also observed for the EC that increased by 28% after compost addition regardless of the sampling time (Table 4 and S3). The percentage of SOM only differed by 1% between the control plots and the plots treated at the first sampling time (11 months after compost
addition). However, after the second compost addition, SOM increased by 23% in the AC amended plots compared to the control. Content of TN improved by 8% with the addition of compost whereas this amount rose up to 20% in the second sampling. Different accumulation patterns of NH4+-N were seen during the two samplings. Thus, the addition of compost caused an increase of 16% at the organic amended plots whereas the control plots showed 2.5 times greater NH4+-N concentrations than those with compost (Table 4 and S3).


**Table 4** Results of ANOVA (F and *p*-value) showing the main significant factors (i.e. compost addition, sampling time or interaction) controlling changes in soil physical and chemical properties at the intensive traditional plots.  SWC: soil water content, EC: electrical conductivity, SOM: soil organic matter, WSC: water-soluble carbon, Avail-K: available K.

| ANOVA results | SWC | | pH | | EC | | SOM | | WSC | | Kjeldahl-N | | NO3⁻-N | | NH₄⁺-N | | Olsen-P | | Avail-K | |
|---|---|---|---|---|---|---|---|---|---|---|---|---|---|---|---|---|---|---|---|---|
| | F | *p* | F | *p* | F | *p* | F | *p* | F | *p* | F | *p* | F | *p* | F | *p* | F | *p* | F | *p* |
| Compost addition | ns | | 12.1* | | 8.14* | | 9.01* | | ns | | ns | | ns | | 5.43* | | ns | | ns | |
| Sampling time | ns | | ns | | ns | | 10.0* | | ns | | 363** | | ns | | 7.21* | | 5.61* | | ns | |
| Compost*sampling time | ns | | ns | | ns | | ns | | ns | | 4.85* | | ns | | ns | | ns | | ns | |


**3.3 Soil enzyme activity in hedgerow grove with irrigation**

Strong seasonal patterns were found with respect to the DHA showing a decrease of the activity of 67% on average irrespective of the treatment from the first sampling to the second (Figure 2, p < 0.05). Compost addition did not seem to have a great effect on this activity but some trends were detected.  During the first sampling, both the NIC and the FC showed an increase of the
DHA of approximately 40% on average, whereas the DC experienced a decrease of 11% compared to its control.  However, for the second sampling the NIC treatment did not show any difference with NI treatment while the compost addition enhanced 40% on average the DHA in the DC and the FC treatments. Irrigation impacted significantly the DHA but only the NI and the F treatments were different (p < 0.05). On average in both years, DHA increased ca. 42% from the NI regime to the F treatment irrespective of the compost addition.




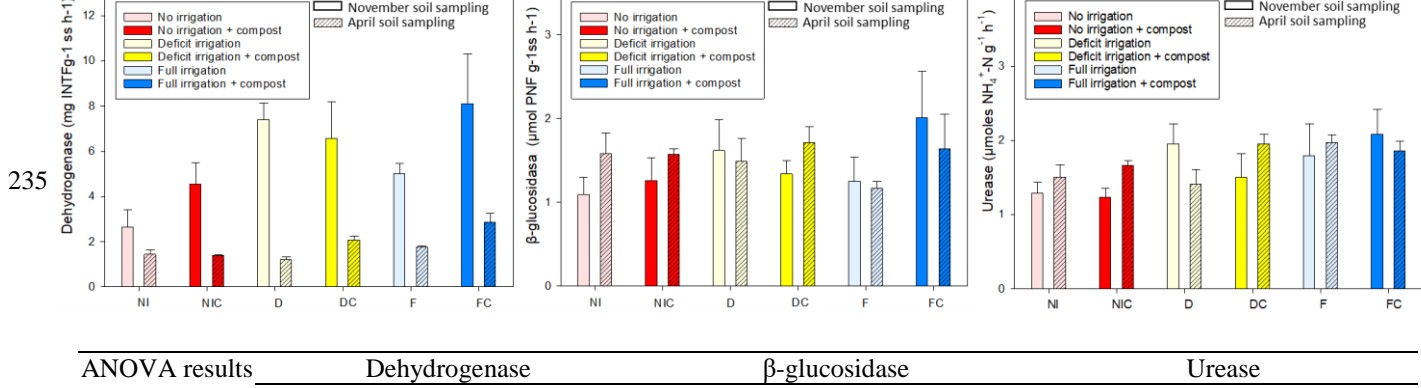

| ANOVA results | Dehydrogenase | | β-glucosidase | | Urease | |
|---|---|---|---|---|---|---|
| | F | *p* | F | *p* | F | *p* |
| Compost addition | ns | | ns | | ns | |
| Irrigation | 4.42 | * | ns | | ns | |
| Sampling time | 96.5 | ** | ns | | ns | |

(* $p < 0.05$, ** $p < 0.001$)

**Figure 2** Mean values of Dehydrogenase (A) β-glucosidase (B) and Urease (C) activities measured for control soils and amended with alperujo compost under different irrigation regimes at the hedgerow plots showing the seasonal changes from November (no shaded bars) to April (shaded bars). Error bars represent standard deviations. No interaction between the Anova factors (compost addition, irrigation regime and sampling time) were found.

The β-Glu and the urease activity did not show any significant difference with respect to the parameters considered (i.e. compost, irrigation regime and sampling time) (Figure 2, $p > 0.05$). As opposed to the DHA, there seemed to be an increase of both activities at the second sampling time except for the D and the FC treatment.

### 3.4 Soil enzyme activity under intensive traditional management and no irrigation

In general, the enzymes activity showed similar patterns as the hedgerow plots, with the DHA displaying a decrease trend in the second soil sampling (Figure 3, $p < 0.05$) whereas the β-Glu and the urease activity tended to increase irrespective of the compost addition (Figure 3). In the first sampling (eleven months after the compost addition), the compost addition did not cause higher enzymatic activities with respect to the control. However, during the second sampling, an increase of 38%, 26% and 31% of DHA, β-Glu and urease respectively was detected with the addition of compost, although this was not significant (Figure 3, $p > 0.05$).



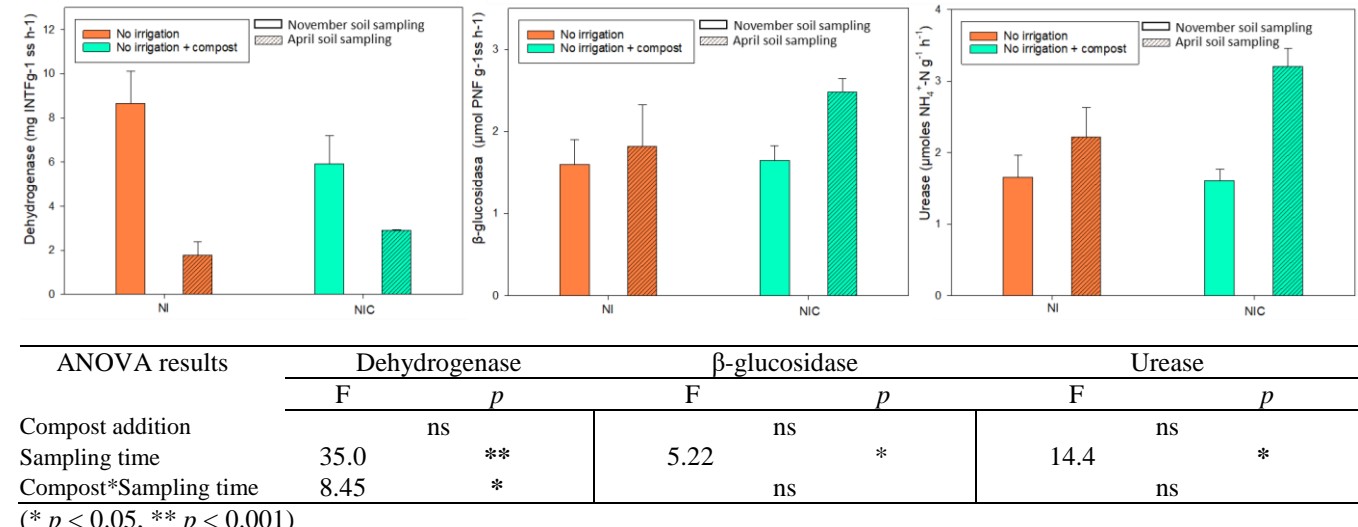

| ANOVA results | Dehydrogenase | | β-glucosidase | | Urease | |
|---|---|---|---|---|---|---|
| | F | *p* | F | *p* | F | *p* |
| Compost addition | ns | | ns | | ns | |
| Sampling time | 35.0 | ** | 5.22 | * | 14.4 | * |
| Compost*Sampling time | 8.45 | * | ns | | ns | |

(* *p* < 0.05, ** *p* < 0.001)

**Figure 3** Mean values of Dehydrogenase (A) β-glucosidase (B) and Urease (C) activities measured for control soil and amended with alperujo compost under a rainfed regime at the intensive traditional plots showing the seasonal changes from November (no shaded bars) to April (shaded bars). Error bars represent standard deviations.

**3.5 Soil physico-chemical correlations and seasonal patterns**

Different pattern of correlations among the variables measured were found at the hedgerow and intensive traditional plots according to the sampling time (Figure 4 and 5). The first sampling time showed the greatest number of correlations at the hedgerow plots, displaying a clear stoichiometric relationship between SOM and the NPK content (p < 0.01). It was also true for the intensive traditional plots although the SOM content was also highly correlated with the β-Glu and the urease activity (Figure 4). Additionally, SWC seemed to exert a great influence on some key chemical parameters (e.g., pH, SOM, inorganic N, TN and enzymes activity) at the intensive traditional plots (with no supplementary irrigation).





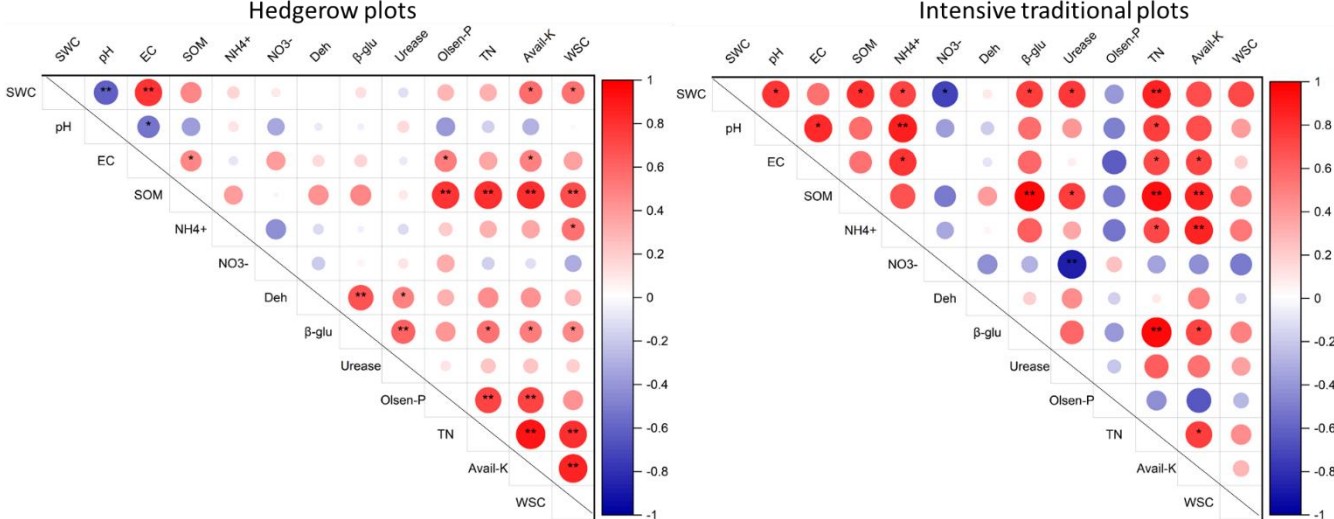

**Figure 4** Pearson's correlation coefficients of soil chemical parameters for the first sampling time (November) at the hedgerow and intensive traditional plots. The circle's color in the correlogram corresponds to the correlation coefficient, wherein a positive correlation (red end of the scale) is closer to 1 and a negative correlation (blue end the of scale) is closer to -1. The size of the circles corresponds to the significance level (* $p < 0.05$, ** $p < 0.01$). Insignificant correlations are not shown. SWC: soil water content, EC: electrical conductivity, SOM: soil organic matter, $NH4^+$: ammonium, $NO_3^-$: nitrate, Deh: dehydrogenase, β-glu: β-glucosidase, TN: total nitrogen, Avail-K: available potassium, WSC: water-soluble carbon.

On the other hand, the second sampling time revealed stronger correlations at the intensive traditional plots. Thus, the SWC, SOM highly correlated with most of the variables and the inorganic N correlated negatively with all enzymatic activities. However, at the hedgerow plots the correlations exhibited among the variables were weak and scarce (Figure 5).

Regarding the seasonal patterns, different rates of soil carbon mineralization between the sampling times were observed and this was dependent on the compost addition and irrigation as well. The percentage of reduction of SOM between the first and the second sampling time was 6% for both NI and NIC treatments and 4%, 25%, 13%, 39% for D, DC, F and FC treatments respectively.



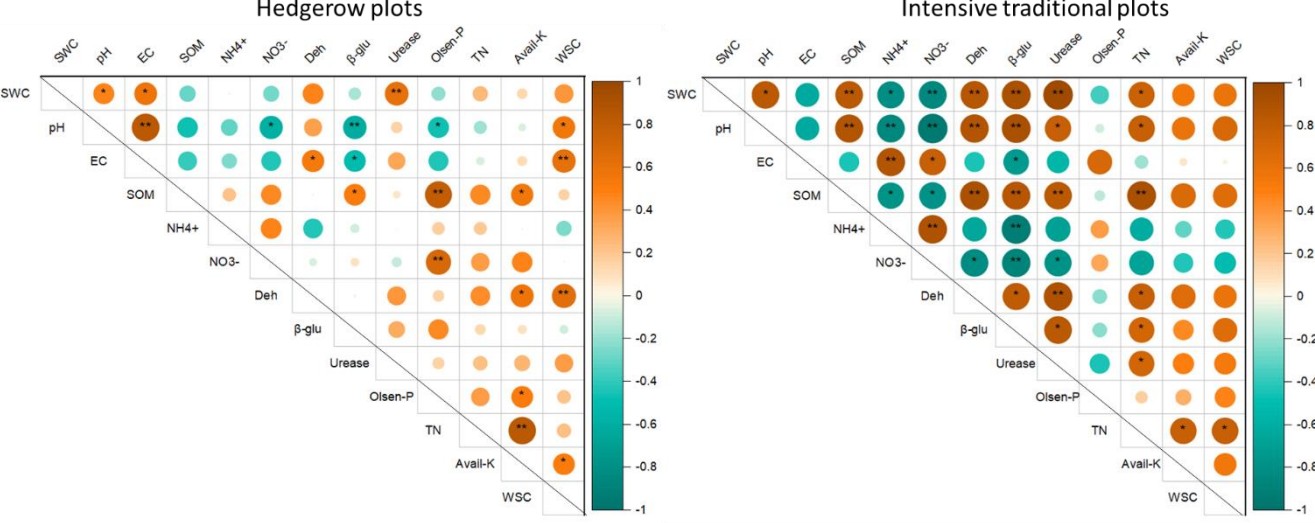

**Figure 5** Pearson's correlation coefficients of soil chemical parameters for the second sampling time (April) at the hedgerow and intensive traditional plots. The circle's color in the correlogram corresponds to the correlation coefficient, wherein a positive correlation (white end of the scale) is closer to 1 and a negative correlation (brown end the of scale) is closer to -1. The size of the circles corresponds to the significance level (* $p < 0.05$, ** $p < 0.01$). Insignificant correlations are not shown. SWC: soil water content, EC: electrical conductivity, SOM: soil organic matter, $NH_4^+$: ammonium, $NO_3^-$: nitrate, Deh: dehydrogenase, β-glu: β-glucosidase, TN: total nitrogen, Avail-K: available potassium, WSC: water-soluble carbon.

## 4. Discussion

### 4.1 Compost and irrigation as drivers of soil properties and enzymatic activities

Intensification of olive groves demands new management strategies to avoid over-fertilization and environmental pollution. Contrasting results are often presented depending on some factors such as the amount of irrigation, tillage practices, nature of the organic amendment and timing of fertilization (Kavvadias et al., 2018b, 2018a; Zipori et al., 2020). In this sense, numerous studies point to irrigation among all these agronomic factors as the pivotal piece driving soil carbon dynamics (Fraga et al., 2020; Pascazio et al., 2018; Sofo et al., 2014). In our case, it is somewhat surprising considering the scarcity of rainfall, that the compost addition had the most striking effect on soil physical-chemical properties, and the irrigation effect was very limited to some soil properties. Likewise, Kavvadias et al. (2018b) could not find a significant effect of the irrigation management on soil properties with the exception of some soil cations concentration (e.g., Mg). However, in their case, they attributed this lack of effect to the fact that their study was based on an area with high precipitations (>950 mm). Pérez et al. (2001) also concluded that irrigation could take up to one year to have a response on vegetative growth and olive grove yield. With our conditions, a much more decisive effect of the irrigation practices would be expected in soil properties. A plausible explanation



could be that the seasonal fluctuation of the sampling times masked the irrigation effect, a decision that was made to test the lasting effect of the compost.

On the other hand, Kavvadias et al. (2018a) found in other study a substantial decrease in soil carbon content in irrigated plots after the addition of the organic amendment compared to the non-irrigated sites. Here, on the contrary, at first, the addition of compost remarkably improved the SOM contents in the full irrigated plots if compared to the non-irrigated but this effect only remained during the first sampling. In the second sampling (just one month after the second compost application) the trend was reversed and we found the lowest SOM content in the full irrigated parcels. Our results suggest that the compost was steadily mineralized at higher moisture contents and once there was a source of carbon available, this was rapidly consumed or leached into deeper soil layers being practically exhausted in the second sampling.

As mentioned in previous studies, a strong relationship between the compost addition and the increase in the availability of N, P and K and SOM content was here observed. Koubouris et al. (2017) reported a markedly improvement in SOM (+30%) following two years of organic amendments and Regni et al. (2017) also reported a positive effect on carbon sequestration after the addition of a mixture of fresh olive pomace deriving from a mixture of a three-phase oil extraction and shredded olive tree pruning residues. Likewise, Fernández-Hernández et al. (2014) detected an improvement of most of the soil characteristics including an increase of SOM content.

Although the compost addition had a clear impact on the SOM and TN, this was not translated in a greater enzymatic response. The DH and β-glu enzymes, which are highly related to the oxidation and hydrolysis of SOM, can be considered as proxies of the intensity of microbial metabolism and as index of soil quality related to the agronomic management (Baležentienė, 2012; Moreno et al., 2009). Numerous studies reported a raise in their activity after the addition of an exogenous source of organic carbon (Federici et al., 2017; Magdich et al., 2020; Panettieri et al., 2022). However, their potential to mineralize organic compounds is often associated with a complex balance between the autochthonous and allochthonous microbial communities that weigh the metabolic investment of the different decomposition pathways of the exogenous and the native source of carbon and are also limited with nutrient and substrate constraints (Lehmann et al., 2020; Panettieri et al., 2022). This balance can be altered in turn by the climatic conditions, irrigation management as in our case or tillage operations. It is well known that irrigation can exert a strong influence on microbial growth and composition and biogeochemical cycles through the control of soil moisture (Bastida et al., 2017; Michalopoulos et al., 2020; Placella et al., 2012) but for us this was mainly reflected in the DHA while the other enzymes were less responsive to this factor.

**4.2 Seasonal influence on soil physico-chemical properties and enzymatic activities**

Understanding the agronomic performance of organic amendments under different irrigation regimes and systems over time is essential to predict future needs and the evolution of the amendments. Here, an overall decay of the compost effect was detected in most of the soil physico-chemical properties over time independently of the irrigation regime or system. In a previous study de Sosa et al. (2021) showed the same seasonal patterns in soil chemical properties following two years of compost application under a traditional intensive management but at higher densities where irrigation is essential, the





synergetic effect of both factors over time should be considered. Little information is available on the combined effects of
irrigation regimes and organic amendments and even less if more intensive or traditional management is considered. Our
results revealed that our source of carbon in combination with water tended to trigger a certain priming effect on the native
SOM with time since the carbon stocks were reduced between 6-38% in the hedgerow plots from the first sampling to the
second as other studies also reported (Kavvadias et al., 2018b). In this sense, it is interesting to note that the deficit irrigation
caused a less intense reduction of the SOM and essential nutrients (i.e. N, P, K) than the full irrigation treatment so it could
represent the best alternative to maximize the agronomics effects of the compost under a water saving strategy. Another aspect
to consider is that carbon stocks did increase over time in the traditional intensive plots with compost amendment so it is clear
that tree density played a key role controlling soil carbon storage and that the amount of compost provided at high densities
was not enough to meet with the crop needs and to promote carbon sequestration. Rui et al. (2016) concluded that the formation
of new biomass increasing soil carbon could be limited by insufficient inorganic nutrients in systems with low inputs and
Ferrara et al. (2015) also observed limited changes in SOM related to a late response of soil quality indicators that could be
the case in our experiment as well.

Our results evidenced that limitations in soil fertility would make necessary a recursive application of compost to maintain
productivity since this type of compost resulted highly biodegradable with time which has been identified before (Panettieri et
al., 2022). However, considering the time factor in the agronomic effect of the compost, it was clear that the deficit irrigation
treatment managed a more efficient compost use by slowing the nutrient loss over time or favouring nutrient storage. For
instance, it is interesting to note that there was an enrichment of P and K contents over time with non-irrigate and deficit
treatments whereas the full irrigation treatment enhanced P mobility as it has been described before (Ibrahimi and Gaddas,
2015; Ojekanmi et al., 2011; Proietti et al., 2015). Given the fundamental role that both macronutrients play for olive trees'
nutrition and in drought tolerance, their evolution should be monitored after successive applications of AC (Christopoulou et
al., 2021; Fernández-Escobar, 2019).

The study of the evolution of the compost over time reflected different strategies of SOM decomposition within the hedgerow
and the intensive plots. During the first sampling SOM highly correlated with NPK content in both management systems which
suggested both processes were coupled after a few months of compost application. However, during the second sampling, right
after the second compost addition, where the soil nutrient status was greatly reduced under hedgerow management, the
correlations among all variables turned weak and difficult to establish. On the contrary, under an intensive traditional
management, a steady carbon assimilation rate mediated by the SWC, the pH and the enzymatic activities continued its course.
No major fluctuations of the enzyme activities were detected over time apart from the DHA that was clearly affected by a
seasonal pattern. Panettieri et al. (2022) identified either greater or no differences of DHA in plots treated with AC according
to the sampling time and the β-glu activity also remained unaltered after the organic amendment. Likewise, Peña et al. (2022)
identified a seasonal response of the DH, β-glu and urease activities after the compost addition whereas Ciadamidaro et al.
(2016) only observed a response of DHA depending on the sampling area and no response for the β-glu activity.





## 5. Conclusions

Agricultural techniques need to be optimized to manage the best compost use and an efficient irrigation management with low
environmental impacts. The application of alperujo compost improved the soil organic matter content in both cultivation
systems. However, the intensive traditional management proved to have a better nutrient balance over time without any
supplemental irrigation. On the other hand, it was clear that when the conversion from traditional to more intensive systems
need to be done, the combined effect of the compost and the irrigation regime has to be taken into account. Our results showed
that the deficit irrigation regime helped to maximize the agronomic effects of the compost and the nutrient supply promoting
in turn a water saving strategy. The full irrigation regime caused a priming effect of the native soil organic matter besides
consuming an amount of water that will surely not be available in a climate change scenario. Moreover, the repeated application
of compost managed a high availability of N, P and K in the soil, effect that tended to disappear under a full irrigation regime.
The sustainability of hedgerow olive groves depends largely on guaranteeing soil fertility in tandem with good water
availability at the most critical stages of the crop and in this sense, the AC in combination with the deficit irrigation proved to
be an efficient tool towards a zero waste circular economy and the water scarcity of climate change.

### Acknowledgments

This study was funded by the Junta de Andalucía within the framework of the project 'Alternative management to ensure the
sustainability of table olive groves in Andalusia' (P20_00492). de Sosa thanks the Junta Andalucía and European Union for
the research grant awarded in the area of the Andalusian Research Development and Innovation (PAIDI 2020). Authors thank
the collaboration of Dr. Girón in the field tasks and Technicians Cristina García, Patricia Puente and Carmen Navarro for their
help in laboratory analysis. This work was framed in the net of experiments of the "Unidad Asociada Uso Sostenible del Suelo
y el Agua en Agricultura".

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
