# Peer review of "Agricultural use of compost under different irrigation strategies in a hedgerow olive grove under Mediterranean conditions. A comparison with traditional systems"

_EGUsphere, 2022_

## Author Response (AR1)

We thank both anonymous reviewers for the supportive and very helpful comments on our manuscript. We have taken on board all the suggestions and modified the manuscript accordingly. You will find below a detailed reply to each of the reviewer's comments. We feel that clarification of the points raised has greatly helped its readability. If further clarification is required on any points, we would be happy to do that.

Main amendments include:

(i) Firstly, we expanded the method section explaining in more detail the compost application and soil sampling as requested by Reviewer #1.

**(ii)** Sedcondly, as requested by Reviewer #2 we have extended the introduction including more references.

**(iii)** Finally, as requested by Reviewer #1, we have avoided the term intensive throughout the manuscript.

**Reviewer 1**

English needs revision, particularly syntaxis needs improvement: e.g. L13-14. Efficient soil and water management is key to ensuring sustainable olive production.
**Response:** A professional language editing service has revised the whole manuscript

L89. How were these 8 plots chosen. Which criteria were used?
**Response:** Plots were chosen to continue with the same treatments (alperujo compost) that had been established in a previous project

L97-105. Explain how (manually, spreader, …) and where the compost is applied with respect to the trees. Was it applied under the tree canopies or in the lanes in-between the tree rows?
**Response:** We added 'the compost based on solid waste from two-phase olive oil extraction (alperujo), was applied at a rate of 17 t ha-1 with a fertilizer spreader in the lanes in between the tree rows in the intensive traditional plots.

L145-148. Explain where the 3 soil samples were taken with respect to the trees and drippers. In the lanes, under the tree canopies, in the tree rows? At what distance from the tree trunks and the drippers? Soil physical, chemical and biological properties are expected to show strong short-range variability in olive groves (e.g. tree rows vs. lanes). Therefore it is important to know exactly where the samples were taken.
**Response:** the following clarification was added: 'Three soil cores (0-10 cm) per plot were taken from the tree rows at approximately 15 cm from the tree trunk and drippers'

L187 EC increased. Is this a consequence of the irrigation water quality? Are measurements of irrigation water quality parameters available? Table S2 and S3, Is EC measured in microS/cm instead of mS/cm?
**Response:** Effectively, we have detected through the periodical analysis of irrigation water a slight accumulation of salts that may be causing this temporary EC increase. This is something that we are monitoring and taking care of to avoid interference with the compost effect. Units of EC in Table S2 and S3 have been corrected.

L287. Use nonsignificant instead of insignificant
**Response:** Done.

L326.Table S1. The deficit irrigation treatment received more water than the full irrigation treatment? Revise. However, this could explain the results discussed in these lines. Maybe it could be interesting to add the irrigation amount for both treatments to the bars in Fig. S1 and add also what is shown in Fig. S2 so that all this information can be interpreted form a single figure. An irregular distribution of rainfall in time could mimic the expected effects of irrigation.
**Response:** Thank you for pointing that out. The irrigation treatments were reversed. We tried to include irrigation amount for all treatments (four new columns) and the result was too messy so we chose to leave it as it was. Nevertheless, we merged Fig. S1 and S2 together as suggested.

Reviewer 2:

the Introduction section seems to be too concise, lacking in sustainability aspects (see line 40) of different planting systems. It can be extended with recent studies, taking into account the environmental impact (CF and WF) of the intensification in olive growing (as 10.3390/su14116389 and 10.1016/j.jclepro.2015.10.088). Moreover, there are some works strictly related to soil management and olive pomace application in hedgerow olive orchard (as 10.1016/j.scienta.2011.04.034 and 10.1016/j.jclepro.2014.06.064 and linked papers) that have to be considered by the Authors. These works could improve the results discussion as well.
**Response:** The introduction section has been extended with the studies provided.

Please use the term traditional and not intensive in the case study reported for a rainfed 286 trees/ha olive orchard, because of intensive is an irrigated orchard with ≥300 trees/ha
**Response:** We replaced the term intensive by traditional.

The trees water status (stem water potential and leaf conductance) data are not reported nor discussed. Please add and appropriately integrate with chemical results.
**Response:** The reason why both parameters are not reported or discussed is because their measurement was not an objective of the study itself nor were their results analyzed but simply they were considered and used as indicators of tree water stress to provide real-time information on soil and plant water status and therefore to adapt the irrigation programs accordingly. The clarification was added in the method section.

 technical corrections

use everywhere the expression cv. Manzanilla instead of Manzanilla cv.
**Response:**Changed.

line 109: revise English language

Response: the section was rewritten.

**Editor:**

Remark 1- "The trees water status (stem water potential and leaf conductance) data are not reported nor discussed. Please add and appropriately integrate with chemical results."

Despite not been the core of the study, the final manuscript will benefit if you could add some basic data on the physiological indicators referred to better clarify the water status of the trees in the experiment.
**Response:** A reference paper with data of the area of study with detailed information of the stem water potential and leaf conductancevalues was added to give further idea of the values obtained in this study.

Remark 2- Reviewers have recommended some additional references in an effort to help you improve your manuscript, which I feel are in the correct direction. Please read and add them in the revised version of the manuscript only if you also consider that they improve your manuscript, but do not feel obliged to add them if you disagree.

**Response:** References have been added.